# Tree-of-Table: Unleashing the Power of LLMs for Enhanced Large-Scale Table Understanding

## Abstract

The ubiquity and value of tables as semi-structured data across various domains necessitate advanced methods for understanding their complexity and vast amounts of information. Despite the impressive capabilities of large language models (LLMs) in advancing the natural language understanding frontier, their application to large-scale tabular data presents significant challenges, specifically regarding table size and complex intricate relationships. Existing works have shown promise with small-scale tables but often flounder when tasked with the complex reasoning required by larger, interconnected tables found in real-world scenarios. To address this gap, we introduce "Tree-of-Table", a novel approach designed to enhance LLMs' reasoning capabilities over large and complex tables. Our method employs Table Condensation and Decomposition to distill and reorganize relevant data into a manageable format, followed by the construction of a hierarchical Table-Tree that facilitates tree-structured reasoning. Through a meticulous Table-Tree Execution process, we systematically unravel the tree-structured reasoning chain to derive the solutions. Experiments across diverse datasets, including WikiTQ, Table-Fact, FeTaQA, and BIRD, demonstrate that Tree-of-Table sets a new benchmark with superior performance, showcasing remarkable efficiency and generalization capabilities in large-scale table reasoning.

## 1 Introduction

Tables, as a pivotal form of semi-structured data, ubiquitously underpin numerous aspects of daily life and professional domains, ranging from open data repositories and web pages to critical applications in financial analysis, risk management, health monitoring, and business reporting (Cafarella et al., 2008). The advent of large language models (LLMs) (OpenAI, 2023; Yao et al., 2023b; Chen, 2023; Jiang et al., 2023; Imani et al., 2023; Anil et al., 2023; Valmeekam et al., 2022) has opened new vistas for understanding and reasoning with tabular data, marking a significant stride in the realm of natural language understanding (Nahid & Rafiei, 2024; Chen et al., 2024; Sui et al., 2024b;a; Ye et al., 2023; Cheng et al., 2022; Jin & Lu, 2023). This intersection is not only instrumental in enhancing the comprehension of tables but also vital for powering a plethora of downstream tasks such as table-based fact verification (Chen et al., 2019) and question answering (Li et al., 2024). Unlike their unstructured text counterparts, tables provide a dense, structured format through the interaction of rows and columns, offering a rich source of information. However, the same structural characteristics pose unique challenges for language models, as they necessitate advanced levels of reasoning over both the textual and numerical data contained within. Given the increasing reliance on tables for data representation and the complexities involved in their interpretation, investigating the integration of LLMs for improved large-scale table understanding has emerged as an essential and compelling research avenue, drawing heightened interest from the global academic and industrial research communities.

Existing methods for table understanding have shown substantial progress in comprehending small-scale tables (Cheng et al., 2022; Ye et al., 2023; Wang et al., 2024). However, these approaches often falter when applied to the more complex and larger tables frequently encountered in real-world scenarios. This gap between academia and practical applications stems from a variety of limitations inherent in current methodologies. One significant challenge is the limited contextual capacity of

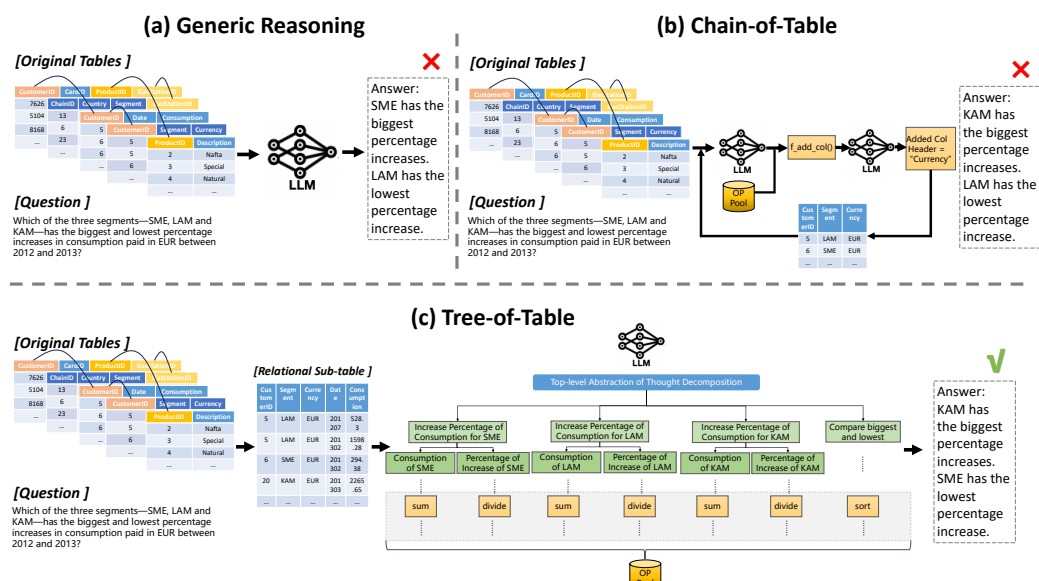

Figure 1: Comparison of (a) Generic Reasoning, (b) Chain-of-Table (Wang et al., 2024), and the proposed (c) Tree-of-Table methods when confronted with large-scale relational tables. Generic Reasoning often struggles with the increased context and complexity, leading to inefficient processing and potential loss of critical information. Chain-of-Table, while more structured with linear thought chain, still faces challenges with the scale and intricacy of data. In contrast, Tree-of-Table showcases a structured and hierarchical reasoning process that adeptly handles large-scale tables, significantly enhancing comprehension and efficiency compared to previous methods, particularly in managing the complexity of expansive tabular data.

today's language models. As tables increase in size, the amount of information that must be processed and understood grows exponentially due to the intricate interactions between rows and columns. This complexity makes it difficult for models to capture and reason about all the necessary information in one go, significantly impeding their understanding capabilities. When faced with complex question-answering logic that spans lengthy chains, pinpointing, extracting, and comprehending key table information becomes an immense challenge.

To address these issues, two main approaches have generally been adopted, as shown in Figure 1. The first involves using only the schema information of tables and employing program-aided methods, such as generating SQL-based answers from questions (Rajkumar et al., 2022b;a; Shi et al., 2020; Pönighaus, 1995; Katsogiannis-Meimarakis & Koutrika, 2023). While this approach avoids directly inputting entire tables, the resulting SQL statements can be lengthy and prone to errors, leading to suboptimal performance. The second strategy involves decomposing tables into multiple sub-tables. Methods like Dater (Ye et al., 2023) attempt to manage larger tables by initially inputting the entire table before breaking it down, which is impractical. The Chain-of-Table (Wang et al., 2024) draws inspiration from the chain-of-thought principle (Wei et al., 2022), performing implicit sub-table extraction. Yet, even this approach is limited to understanding smaller tables. Additionally, traditional table understanding datasets like WikiTQ (Pasupat & Liang, 2015) and TableFact (Chen et al., 2019), which are relatively small, severely restrict the exploration of large-scale table understanding. Fortunately, the introduction of the BIRD (Li et al., 2024) dataset, considered the largest and most complex table understanding dataset to date, highlights the pressing need for improvements. Despite this, due to the reasons mentioned above, existing large language models still exhibit low accuracy on comprehensive, large-scale table datasets like BIRD (Li et al., 2024), signaling a clear necessity for methodological innovations in this area.

Addressing these concerns, we propose "Tree-of-Table", a novel paradigm crafted to optimize LLMs for the task of large-scale table understanding, as shown in Figure 1. By condensing and decomposing tables, our approach distills and systematizes the critical information into a tree-structured model that resonates with the stepwise reasoning employed by humans. This tree acts as a roadmap,

guiding the LLM through the complexities of the table in a logical and organized manner. It provides a structured approach where each node serves a purpose, simplifying the interaction between the LLM and the tabular data. The efficacy of our Tree-of-Table methodology is emphatically validated through rigorous testing across a selection of datasets (including WikiTQ (Pasupat & Liang, 2015), TableFact (Chen et al., 2019), FeTaQA (Nan et al., 2022), and BIRD (Li et al., 2024)) with each presenting its own unique challenges. Consistently achieving top-tier results, Tree-of-Table demonstrates not just its capacity to navigate the intricacies of table reasoning but also its potential to set a new benchmark in the field.

## 2 RELATED WORK

**Traditional Table Understanding.** At the core, traditional methods have focused on generating executable languages like SQL (Rajkumar et al., 2022b; Liu et al., 2021; Eisenschlos et al., 2020; Jiang et al., 2022) to interact with tables. This approach stems from the need to reason over both free-form natural language questions and (semi-)structured tables. While effective in accessing tabular data, these methods often fall short in capturing the nuanced semantics within a table, particularly struggling with web tables that feature free-form text in cells.

**Prompting Language Models for Table Understanding.** A novel stride in table understanding has been the application of prompting strategies (Wei et al., 2022; Chen et al., 2022; OpenAI, 2023; Imani et al., 2023; Khot et al., 2022; Zhang et al., 2022). By generating reasoning steps through in-context learning, models like Chain-of-Thought (Wei et al., 2022) and its evolutions (Yao et al., 2023a) break down questions into sub-problems, iteratively solving each to improve comprehension of complex tasks. These methods showcase LLMs' prowess in handling intricate reasoning chains, albeit not being explicitly designed for tabular data. Emerging approaches have sought to extend LLM capabilities beyond text, incorporating external tools to solve reasoning tasks (Cheng et al., 2022; Hsieh et al., 2023; Dhingra et al., 2019; Liu et al., 2023). Generating Python or SQL programs (Cheng et al., 2022; Nahid & Rafiei, 2024; Shi et al., 2020; Pönighaus, 1995) and executing them with interpreters or APIs has shown promise in enhancing arithmetic and table-based reasoning. However, the performance of these program-aided methods sometimes falters in complex table scenarios due to the static nature of tables in the reasoning process. Dater (Ye et al., 2023) dynamically modifies the tabular context to aid in solving table-based tasks, albeit primarily focusing on data pre-processing with limited operations. Subsequently, the Chain-of-Table (Wang et al., 2024) method is inspired by the chain-of-thought (Wei et al., 2022) principle and performs implicit sub-table extraction. Contrarily, our proposed Tree-of-Table approach is inspired by the tree-of-thought principle (Yao et al., 2023a), creating adaptive tree-based reasoning chains that exploit the planning capabilities of LLMs for more nuanced and context-specific table reasoning.

**Table Understanding Datasets.** Datasets like WikiTQ (Pasupat & Liang, 2015) and TableFact (Chen et al., 2019) have been instrumental in developing table understanding methods. These standard benchmarks provide a foundation but are often limited in size and complexity. BIRD (Li et al., 2024) represents a significant leap forward in the field, being one of the largest and most intricate datasets designed for table understanding to date. Spanning across 37 professional domains with a substantial size of 33.4 GB, BIRD offers over 12,000 examples gleaned from real-world databases. Its development involved modifying open-source relational databases and curating additional ones, all complemented by crowdsourced natural language questions and corresponding SQL queries.

## 3 TREE-OF-TABLE: UNLEASHING THE POWER OF LLMS

### 3.1 FORMULATION OF LARGE-SCALE TABLE UNDERSTANDING

In the domain of table understanding, the core challenge lies in accurately interpreting and extracting information from tabular data in response to a given natural language query or statement. The essence of table understanding can be encapsulated as the task of mapping a natural language question or statement $Q$ to a corresponding output $S$ that accurately reflects the information contained within a table $T$. This table can be characterized by its structure, which includes rows and columns, with each cell representing a specific data point. Formally, a table $T$ can be divided into headers $H$ and

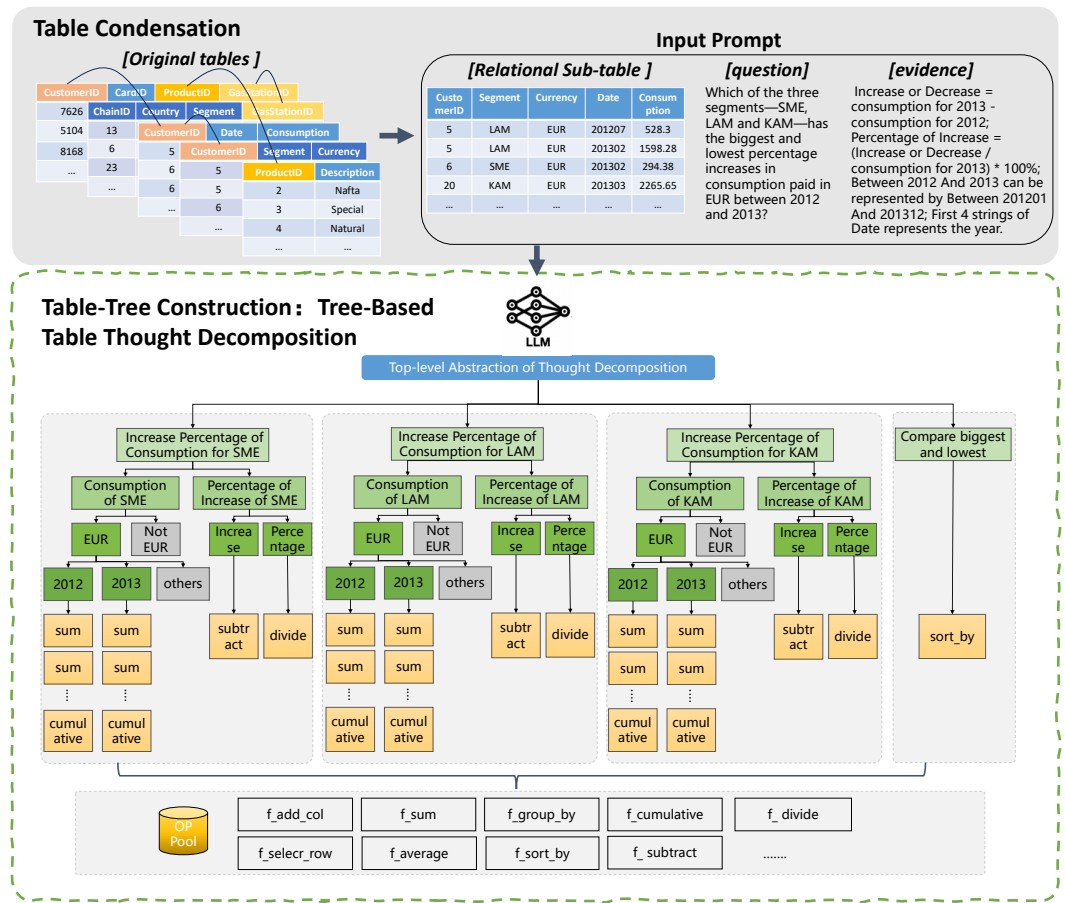

Figure 2: Illustration of the initial phases in the Tree-of-Table methodology, encompassing Table Condensation (the upper part), followed by Table-Tree Construction (the lower part). Starting with a large-scale input table, the process selectively condenses the data, emphasizing task-relevant information. Subsequently, the decomposed elements are methodically reorganized into a Table-Tree, a hierarchical structure designed to streamline and guide the subsequent reasoning process.

data values $D$, where each header in $H$ corresponds to a column in the table and $D$ represents the collective data points contained within these columns.

Furthermore, table understanding involves not just the direct interpretation of tables but also potentially requires external knowledge $K$ and conversion of table data into a format that is amenable to computational models. This is especially relevant for tasks that involve complex reasoning or necessitate an understanding beyond the explicit table content, such as requiring background knowledge or contextual understanding to correctly interpret the question or the data.

In our experiments, we utilize a total of four datasets: WikiTQ (Pasupat & Liang, 2015), TabFact (Chen et al., 2019), FeTaQA (Nan et al., 2022), and BIRD (Li et al., 2024). For WikiTQ, TabFact, and FeTaQA, there is no external knowledge; therefore, the table understanding problem can be defined as finding a function or model $f(\cdot, \theta)$ that satisfies

$$S = f(Q, \langle H, D \rangle | \theta), \tag{1}$$

where $\theta$ represents the model parameters. In contrast, for the BIRD dataset, there is external knowledge used to explain specific terms in the questions, allowing us to define the table understanding problem as

$$S = f(Q, \langle H, D \rangle, K | \theta), \tag{2}$$

where $K$ denotes the external knowledge.

## 3.2 OVERVIEW

In this work, we introduce a novel approach named "Tree-of-Table" devised to address the challenge of table reasoning within large-scale table understanding datasets (e.g., BIRD) and real-world applications, as shown in Figure 2 and Figure 3. Our methodology encompasses several steps designed to simplify and enhance the reasoning capabilities of LLMs when confronted with large, interconnected tables. First, we condense the tables based on the specific requirements of the query. This process identifies relevant portions of the tables, thereby reducing the cognitive load on LLMs. We then apply a tree-based decomposition strategy to segment large tables into smaller, manageable units, guided by the relationships among tables, such as foreign keys, and the structure of the query. Next, we construct a "Table-Tree" by reorganizing the condensed information into a hierarchical structure. Each node in this tree represents a logical block of information or a step in the reasoning process, mirroring the cognitive approach of breaking down complex problems into simpler sub-problems. Finally, we perform a sequential traversal of the constructed Table-Tree to derive answers to the queries. This systematic traversal ensures logical progression through each node, allowing for the synthesis of information and insights gained from previous steps. The iterative nature of this reasoning process culminates in a well-informed conclusion.

## 3.3 TABLE CONDENSATION AND DECOMPOSITION

Addressing the significant challenges posed by large-scale relational tables to LLMs requires a nuanced understanding of the specific difficulties in question. These challenges primarily stem from two aspects: (1) The intricate foreign key relationships among multiple tables, which are commonly defined using SQL syntax, may not be readily interpretable by LLMs due to their complexity and the specialized knowledge required to understand relational database schemas. (2) The sheer size of the tables often exceeds the input context limit of LLMs, making it impossible for these models to process the entirety of the data directly. To effectively address these issues, our methodology incorporates two key processes: Table Condensation and Tree-based Decomposition.

### 3.3.1 TABLE CONDENSATION

As shown in the upper part of Figure 2, our initial step involves condensing the tables based on the context of the question $Q$ and any additional evidence provided. Since there are possibly multiple tables, this process employs LLMs to identify one sub-table relevant to $Q$ from them through schema-linking (Lei et al., 2020). Following the identification of relevant schemas/headers, we merge these multiple tables to reduce redundancy and decrease their size,

$$\text{subTable}_Q = f(\text{Schema\_Link}(Q, \{H\}, K)|\theta), \tag{3}$$

where $\{H\}$ indicates the headers of the tables. This condensation aims to recall one sub-table pertinent to $Q$ and eliminate superfluous table information, thereby enhancing the information density related to $Q$ within the tables and making it more manageable for LLM processing. Through a detailed analysis of the BIRD dataset, we observed that over 70% of questions involved tables whose length exceeded the input limitations of current LLMs. Furthermore, more than 90% of these questions pertained to at least two tables, with 20% involving four or more tables, significantly complicating the understanding process for LLMs. Post-condensation, we found that the length of tables involved in more than 60% of long questions was reduced below the LLM input limit, and all questions were associated with a singular, condensed table, as shown in the upper part in Figure 2.

### 3.3.2 TREE-BASED DECOMPOSITION

Even with reduced size and complexity post-condensation, the tables might still be too lengthy or intricate for LLMs to handle efficiently, occasionally still surpassing the models' input limits. To mitigate this, we begin by breaking down the question $Q$ into its most general components, delineating the entire problem-solving process into several independent yet sequentially connected steps.

$$S = \mathcal{P}_{\text{decomp}}(\{S_i^1\}, r^1|Q), \quad i < \text{MAXDegree}, \tag{4}$$

where $S$ is the final solution, $\{S_i^1\}$ is the firstly decomposed intermediate sub-solutions towards $S$, $r^1$ is the possible relationship between $\{S_i^1\}$. $\mathcal{P}_{\text{decomp}}$ is the "Thought Decomposition Prompt". "MAXDegree" is the pre-defined maximum degree of the Table-Tree. This decomposition involves

mapping out the key stages ($\{S_i^1\}$ and $r^1$) of reasoning required to address $Q$. By doing so, we transform a potentially overwhelming task into a series of manageable sub-tasks, each contributing incrementally to the formulation of the final answer. $\{S_i^1\}$ and $r^1$ also serve as the root node of the first-level subtree we will construct in the Table-Tree, for example, as shown the lower part in Figure 2.

## 3.4 TABLE-TREE CONSTRUCTION

Drawing inspiration from the Tree-of-Thought concept (Yao et al., 2023a), our table-tree structure closely resembles how humans naturally approach problem-solving. The illustration of overall construction is showed in the lower part in Figure 2. When faced with a complex problem, people typically employ a "breadth-first" strategy: deconstructing the problem into several general, independent yet interconnected subprocesses and then iteratively refining each subprocess into finer-grained solutions.

### 3.4.1 BREADTH-FIRST THOUGHT GENERATION

Within this framework, we utilize in-context learning to instruct LLMs on dynamically generating thoughts for the question in a breadth-first way. Based on the firstly decomposed $\{S_i^1\}$ and $r^1$ in Eq. 4, the following breadth-first thought generation process can be formulated as,

$$
\begin{aligned}
S &= \mathcal{P}_{\text{decomp}}(\{S_i^1\}, r^1 | Q), \quad i < \text{MAXDegree}, \\
S_i^1 &= \mathcal{P}_{\text{decomp}}(\{S_{i,j}^2\}, r_j^2), \quad j < \text{MAXDegree}, \\
&\quad\quad\quad\quad ...... \\
S_{i,j,...}^d &= \mathcal{P}_{\text{decomp}}(\{S_{i,j,...,k}^{d+1}\}, r_{i,j,...,k}^{d+1}), \quad k < \text{MAXDegree}, \\
&\quad\quad\quad\quad ...... \\
S_{i,j,...,k,...}^{d_{\max}-1} &= \mathcal{P}_{\text{decomp}}(\{S_{i,j,...,k,...,l}^{d_{\max}}\}, r_{i,j,...,k,...,l}^{d_{\max}}), \quad d_{max} <= \text{MAXDepth},
\end{aligned}
\tag{5}
$$

where $S$ is the final solution, $\{S_i^1\}$ is the firstly decomposed intermediate solutions towards $S$, $r^1$ is the possible relationship between $\{S_i^1\}$. $\{S_{i,j,...,k}^{d+1}\}$, $r_{i,j,...,k}^{d+1}$, and so on. Note that $r_{i,j,...,k}^{d+1}$ may be empty in the actual decomposition process if we need not to consider the relationship between $\{S_{i,j,...,k}^{d+1}\}$. $d$ is the depth of thought. "MAXDegree" and "MAXDepth" are the pre-defined maximum degree and depth of the Table-Tree, respectively.

To prevent excessive decomposition of thoughts that could lead to redundant or erroneous reasoning processes, we set a maximum value for the depth $d$, denoted as "MAXDepth", and follow (Yao et al., 2023a) to utilize the LM to deliberately reason about end thought states $\{S_{i,j,...,k,...,l}^{d_{\max}}\}, r_{i,j,...,k,...,l}^{d_{\max}}$. Such a deliberate heuristic can be more flexible than programmed rules.

### 3.4.2 ITERATIVE CONSTRUCTION

Building upon the breadth-first thought generation, we construct the Table-Tree by iteratively constructing child nodes level by level for $\{S_i^1\}$ and $r^1$, until we reach the leaf nodes at the bottom of the tree. Each sub-thought corresponding to $\{S_{i,j,...,k}^{d+1}\}$ and $r_{i,j,...,k}^{d+1}$ is regard as intermediate node, which act as "thinking node" representing a subprocess that can be further decomposed. The end thought state $\{S_{i,j,...,k,...,l}^{d_{\max}}\}, r_{i,j,...,k,...,l}^{d_{\max}}$ are set to leaf nodes, which functions as "execution nodes". In concrete, leaf nodes are the actionable endpoints of Table-Tree where specific operations such as data retrieval, calculations, or logical evaluations occur based on the parameters defined by their parent nodes. Therefore, we formulate $\{S_{i,j,...,k,...,l}^{d_{\max}}\}, r_{i,j,...,k,...,l}^{d_{\max}}$ as

$$
S_{i,j,...,k,...,l}^{d_{\max}}, r_{i,j,...,k,...,l}^{d_{\max}} = \mathcal{P}_{\text{sample}}(\{\text{OP\_Pool}\}), \quad d_{max} <= \text{MAXDepth}.
\tag{6}
$$

where $\mathcal{P}_{\text{sample}}$ is the "Operation Sample Prompt". In selecting the operation pool, we based it on (Wang et al., 2024) and chose the most frequently used table operations from the resource at (Bytescout, 2024).

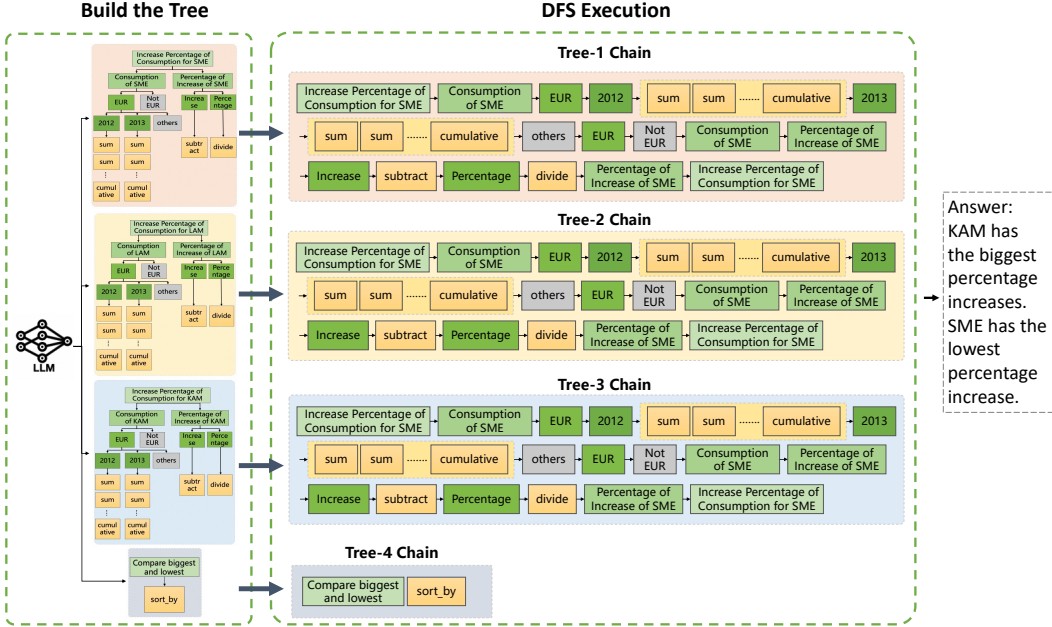

Figure 3: Depiction of the Table-Tree Execution phase within the Tree-of-Table approach. The model traverses the hierarchical Table-Tree, processing each node sequentially from the root to the leaves. At each step, the model integrates the information from the current node with the insights gathered from previous nodes, systematically building upon the reasoning chain to derive the final answer.

## 3.5 TABLE-TREE EXECUTION

After constructing the Table-Tree, we view it as a proxy task for the entire table understanding procedure. As shown in Figure 3, by traversing and executing operations across this tree, LLMs can implicitly generate tables and save intermediary results, thus enabling a seamless reasoning process. This stage diverges from the construction phase by utilizing a depth-first search approach to execute the thought chain, ensuring a systematic and comprehensive exploration of the tree structure.

**Depth-First Search Execution.** The rationale behind adopting a depth-first search (DFS) (Tarjan, 1972) strategy for Tree Execution lies in its ability to fully explore and resolve each branch of the tree to completion before moving to the next. This method aligns with the logical progression of solving a complex problem by focusing on and completing one aspect of the problem entirely, ensuring that all necessary computations and logical deductions related to a branch are performed before considering alternative or subsequent branches.

**Leveraging Tree Structure for Efficiency.** A key advantage of the tree structure is its inherent ability to enhance reasoning efficiency by logically organizing and compartmentalizing different aspects of the problem-solving process into subtrees. To exploit this benefit to its fullest, we execute the reasoning process subtree by subtree, based on the root node's children. After processing a subtree, we store its result before proceeding. This approach contrasts with linearly merging all subtrees into a single chain for execution. By maintaining the distinction between subtrees and executing them as separate units, we significantly mitigate the risk of intermediary tables becoming excessively large and unwieldy, which in turn, would thwart the reasoning process.

## 3.6 COMPARISON WITH CHAIN-OF-TABLE

In summary, the differences between the two methods are as follows: (1) Planning Style: Tree-of-Table uses a hierarchical, tree-based thought decomposition approach, while Chain-of-Table uses a linear thought decomposition approach. (2) Execution Strategy: Notably, Chain-of-Table processes the entire chain of history for each dynamic planning step and is shown to be effective for relatively small tables, such as WikiTQ. However, as tables grow larger and questions become more complex,

Table 1: Comparison of Table Understanding results on WikiTQ, TabFact datasets, with GPT3.5, PaLM2 and LLaMA2.

| Method | WikiTQ | | | TabFact | | |
|---|---|---|---|---|---|---|
| | GPT3.5 | PaLM2 | LLaMA2 | GPT3.5 | PaLM2 | LLaMA2 |
| Text-to-SQL (Rajkumar et al., 2022b) | 52.90 | 52.42 | 36.14 | 64.71 | 68.37 | 64.03 |
| End-to-End QA (Wang et al., 2024) | 51.84 | 60.59 | 23.90 | 70.45 | 77.92 | 44.86 |
| Few-Shot QA (Wang et al., 2024) | 52.56 | 60.33 | 35.52 | 71.54 | 78.06 | 62.01 |
| Binder (Cheng et al., 2022) | 56.74 | 54.88 | 30.92 | 79.17 | 76.98 | 62.76 |
| Chain-of-Thought (Wang et al., 2024) | 53.48 | 60.43 | 36.05 | 65.37 | 79.05 | 60.52 |
| Dater (Ye et al., 2023) | 52.81 | 61.48 | 41.44 | 78.01 | 84.63 | 65.12 |
| Chain-of-Table (Wang et al., 2024) | 59.94 | 67.31 | 42.61 | 80.20 | 86.61 | 67.24 |
| **TREE-OF-TABLE** | **61.11** | **68.77** | **44.01** | **81.92** | **87.88** | **69.33** |

Table 2: Comparison of Table Understanding results on FetaQA and BIRD datasets.

| Method | FeTaQA | | | | BIRD | | | |
|---|---|---|---|---|---|---|---|---|
| | BLEU | ROUGE-1 | ROUGE-2 | ROUGE-L | BLEU | ROUGE-1 | ROUGE-2 | ROUGE-L |
| FT(T5-large) (Ye et al., 2023) | 30.54 | 0.63 | 0.41 | 0.53 | - | - | - | - |
| End-to-End QA (Wang et al., 2024) | 28.37 | 0.63 | 0.41 | 0.53 | 9.90 | 0.44 | 0.18 | 0.43 |
| Codex (Chen et al., 2021) | 27.96 | 0.62 | 0.40 | 0.52 | - | - | - | - |
| Dater (Ye et al., 2023) | 29.47 | 0.63 | 0.41 | 0.53 | 10.65 | 0.44 | 0.18 | 0.43 |
| Chain-of-Table (Wang et al., 2024) | 32.61 | 0.66 | 0.44 | 0.56 | 12.12 | 0.49 | 0.22 | 0.48 |
| **TREE-OF-TABLE** | **34.73** | **0.68** | **0.46** | **0.58** | **15.70** | **0.53** | **0.26** | **0.52** |

maintaining the complete thought chain becomes cumbersome, ultimately decreasing the efficiency of the model. Our Tree-of-Table method addresses this by embracing the tree's inherent "divide and conquer" philosophy (Bentley, 1980) to construct a Table-Tree. Each generation of child nodes relies exclusively on the information from their multi-level parent nodes, without the need for uncle nodes, as illustrated in Figure 2. By this way, we significantly reduce the historical chain's length on which each node's dynamic planning relies, to less than the depth of the tree, thus considerably simplifying the generation process at each level. (3) Data Preprocessing: Chain-of-Table does not preprocess complex tables with rich foreign key connections, whereas Tree-of-Table includes a Table Condensation step to handle such complexities.

# 4 EXPERIMENTS

## 4.1 DATASETS, METRICS, IMPLEMENTATION DETAILS

We evaluate our method on both three small table understanding benchmarks: WikiTQ (Pasupat & Liang, 2015), FeTaQA (Nan et al., 2022), and TabFact (Chen et al., 2019), and one large-scale dataset: BIRD (Li et al., 2024). For WikiTQ and TabFact, we employ the standard denotation accuracy metric. The nature of FeTaQA and BIRD for requiring elaborate responses prompts us to assess performance through a variety of metrics including BLEU, ROUGE-1, ROUGE-2, and ROUGE-L (Lin, 2004) to capture different facets of response quality. For our experiments, following the previous works (Wang et al., 2024), we leverage the computational prowess of advanced language models, namely PaLM 2 (Anil et al., 2023), GPT 3.5 (OpenAI, 2023), and LLaMA 2 (Touvron et al., 2023). To facilitate in-context learning, we incorporate a few-shot approach using demo samples from the training set within the prompts, ensuring the models can effectively learn from limited examples.

## 4.2 MAIN RESULTS

In our experimental evaluation, we comprehensively compare our proposed approach, Tree-of-Table, with several renowned baselines and state-of-the-art methodologies across both small and large-scale tabular datasets, including WikiTQ, TableFact, FeTaQA, and BIRD, in Table 1 and Table 2. Our analysis is designed to assess the effectiveness of Tree-of-Table in facilitating complex table understanding and reasoning tasks, particularly highlighting its performance in challenging scenarios involving large-scale tables. The results in Table 1, demonstrate that Tree-of-Table not only significantly outperforms early Generic Reasoning methods (End-to-End QA, Few-Shot QA, Chain-of-Thought) and Program-aid Reasoning methods (Text-to-SQL, Dater, Binder) but also surpasses

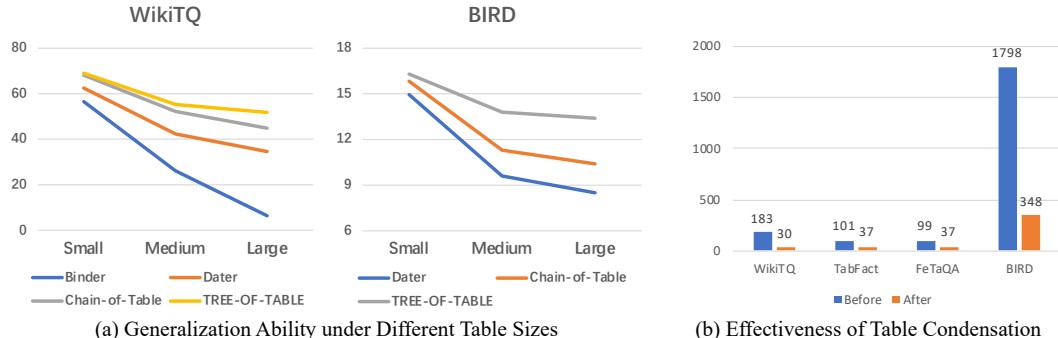

(a) Generalization Ability under Different Table Sizes     (b) Effectiveness of Table Condensation

Figure 4: Ablation study: (a) Generalization Ability under Different Table Sizes. (b) Effectiveness of Table Condensation.

the current state-of-the-art method, Chain-of-Table. This superiority was consistent across multiple LLMs including GPT3.5, PalM2, and LLAMA2. Our Tree-of-Table approach showcased robust enhancements in reasoning over both small and large tables. Specifically, in datasets with larger tables like BIRD, the benefits of Tree-of-Table were even more pronounced, suggesting that our tree-based method is particularly suited for complex, large-scale reasoning tasks where effective condensation, decomposition, and subtree execution strategies are critical.

## 4.3 ABLATION STUDY

**Generalization Ability under Different Table Sizes.** Here, we evaluate the generalization capacity of our method across tables of varying sizes. Large tables present considerable comprehension challenges to models, as their capacity to grapple with extended contexts and prompts is severely tested. In Figure 4, we provide a detailed comparison of 4 methodologies (Binder, Dater, Chain-of-Table, and Tree-of-Table) across two datasets (WikiTQ and BIRD). The performance metrics in table understanding tasks evidently deteriorate as the size of the tables increases. This degradation in performance reflects the inherent difficulties associated with large table comprehension, confirming that it remains an exceptionally challenging problem area.

However, it is notable that the decline in performance with increasing table size is much more gradual for Tree-of-Table as compared to other methods. Especially on the large-scale table dataset BIRD, Tree-of-Table demonstrates superior robustness and generalization ability. Even as table size scaled up, Tree-of-Table maintaines a level of performance that was not only

Table 3: The Node Number and Height of Tree Chains.

| Dataset | WikiTQ | TabFact | BIRD |
|---|---|---|---|
| Chain-of-Table: Chain Length | 4 | 4 | 11 |
| Tree-of-Table: Tree Height | 3 | 3 | 7 |
| Tree-of-Table: Node Number | 6 | 7 | 18 |

better than its counterparts but also displayed less variance in its results.

**Node Number and Height of Tree Chains.** The height of the Table-Tree reflects the depth of the model's reasoning chain, indicative of the complexity of the reasoning process. In contrast, the node number corresponds to the length of the thought chain, representing the number of discrete reasoning steps taken by the model. In Table 3, our comparative analysis reveals that across both smaller and larger datasets, the average height of the Table-Trees is generally less than that of the Chain-of-Table. This indicates that the reasoning process in the Tree-of-Table method tends to require fewer levels of hierarchical reasoning to arrive at a solution. Additionally, the average length of the Table-Trees remains within a reasonable range, suggesting a good balance between depth and breadth in our tree structures.

**Comparison of Table Format Encoding.** The encoding format of tables plays a vital role in how effectively a model can interpret and manipulate table data. Early research has indicated that the specific form of table encoding can significantly impact the model's performance in table understanding tasks. Here, we follow the lead of prior work, comparing the effects of four distinct

encoding formats on the final performance of table understanding: PIPE, HTML, TSV (Tab Separated Values), and Markdown. As shown in Table 4, the Markdown format leads to the highest performance among the tested encoding styles. The benefits of Markdown may be likely attributed to its readability, clear structure, and straightforward syntax, all of which align well with the parsing capabilities of LLMs.

**Efficiency Analysis.** In this context, efficiency refers to the model's capability to achieve its goals with the least amount of computational resources—specifically, the number of samples it needs to generate to arrive at a correct answer. To substantiate the efficiency of our Tree-of-Table methodology, we scrutinize how it compares with existing methods in terms of the number of required generated samples to solve tasks. For a comprehensive analysis, we compared Tree-of-Table against notable methods such as Dater and Chain-of-Table on BIRD. As depicted in Table 5, our analysis demonstrates that Tree-of-Table consistently requires the fewest generated samples to reach accurate answers across all evaluated datasets.

Table 4: Comparison of Table Format Encoding.

| Table Formatting | WikiTQ |
| --- | --- |
| HTML | 68.01 |
| TSV | 68.12 |
| PIPE | 69.34 |
| MarkDown | 69.77 |

Table 5: Efficiency Analysis.

| Method | Generate Samples |
| --- | --- |
| Dater | 300 |
| Chain-of-Table | 120 |
| TREE-OF-TBALE | 90 |

**Effectiveness of Table Condensation.** Finaly, we validate the efficacy of the proposed Table Condensation component in reducing table sizes, making them more amenable to LLMs for reasoning. By condensing tables, we aim to filter out irrelevant information, thereby boosting the signal-to-noise ratio and allowing the model to focus on the most pertinent data. We conducte a comparative analysis of the number of table cells before and after applying Table Condensation across four datasets: WikiTQ, Tab-Fact, FeTaQA, and BIRD. Figure 4 (b) highlights the stark contrast in table sizes before and after the application of Table Condensation. Across all examined datasets, there is a significant reduction in the number of table cells post-condensation. This reduction demonstrates the effectiveness of our method in shrinking table dimensions, ensuring that tables remain within a tractable size range and contain information that is highly relevant to the task at hand.

## 5 CONCLUSION

In this paper, we address the profound challenge of advancing table understanding with LLMs, specifically in the domain of large and complex tabular datasets. Our innovative approach, Tree-of-Table, integrates table condensation and decomposition with a hierarchical reasoning construct that aligns with human cognitive processes to tackle intricate problem-solving tasks. Our extensive experiments conduct across various datasets, including WikiTQ, TableFact, FeTaQA, and BIRD, demonstrate that Tree-of-Table not only achieves state-of-the-art performance but also presents remarkable improvements in efficiency and generalizability.

**Limitations.** While Tree-of-Table has shown exceptional results in enhancing the efficiency and effectiveness of large language models in processing extensive tabular data, the tree-based reasoning may need to require careful calibration to balance depth and breadth effectively—a task that necessitates fine-tuning and may impose certain limitations on adaptability.

**Broader Impact.** The broader impact of Tree-of-Table is multi-faceted, extending across academic, industrial, and societal domains. Academically, our work contributes a significant leap forward in the intersection of table understanding and natural language processing, providing a reference point for future research and development in this area. In industry, the application of Tree-of-Table can revolutionize the way organizations interact with large datasets. By simplifying the complexity and enhancing the reasoning capabilities of models with tabular data, Tree-of-Table can facilitate more informed decision-making, enhance prediction systems, and optimize data-driven strategies across various sectors such as finance, healthcare, and logistics. From a societal perspective, improving the accessibility and comprehension of large-scale data has the potential to democratize information. By enabling a more nuanced understanding of data presented in tabular form, Tree-of-Table can contribute to greater transparency and empower individuals to make better data-informed decisions.

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

## A  Appendix

### A.1  More Experiments

#### A.1.1  Ablation study on time costs

In addition to the existing efficiency analysis based on the number of samples (in Sec 3.3, Table 5), we further include a time cost analysis in Table 6, which shows the comparison of overall time costs between Tree-of-Table and the current state-of-the-art, Chain-of-Table.

Table 6: Comparison of overall time costs between Tree-of-Table and Chain-of-Table.

| Method | Chain-of-Table | Tree-of-Table |
|---|---|---|
| Time Cost | 5.7s | 7.8s |

Table 7: Accuracy evaluation results for Tree-of-Table in comparison with other Text-to-SQL methods.

| Methods | DIN-SQL | MAC-SQL | DTS-SQL | MCS-SQL | Tree-of-Table |
|---|---|---|---|---|---|
| Accuracy (%) | 50.72 | 57.56 | 55.80 | 63.36 | 65.07 |

Table 8: Selection of MAXDepth.

| MAXDepth | 6 | 8 | 10 |
|---|---|---|---|
| Accuracy(%) | 59.96 | 61.11 | 60.47 |

Table 9: Selection of MaxDegree.

| MAXDegree | 3 | 4 | 5 |
|---|---|---|---|
| Accuracy(%) | 60.03 | 61.11 | 60.91 |

Table 10: Error Propagation

| Remove Percent(%) | 5 | 8 | 10 | 12 | 15 |
|---|---|---|---|---|---|
| Accuracy(%) | 61.09 | 61.00 | 60.87 | 60.22 | 59.45 |

#### A.1.2  Accuracy evaluation results

In addition to Table 2, we also show the accuracy evaluation results for Tree-of-Table in comparison with other Text-to-SQL methods such as DIN-SQL (Pourreza & Rafiei, 2024a), MAC-SQL (Wang et al., 2023), DTS-SQL (Pourreza & Rafiei, 2024b), and MCS-SQL (Lee et al., 2024), in Table 7.

#### A.1.3  Selection of MAXDegree and MAXDepth

In our experiments, we initially conduct preliminary experiments to roughly determine their value ranges. We then perform detailed hyperparameter experiments to identify the optimal values. The ablation study results on WikiTQ are listed in the Table 9 and Table 8. It can be seen that our proposed method is relatively robust with respect to these two hyperparameters.

#### A.1.4  Error Propagation

We also conduct an error propagation study on WikiTQ in Table 10. Specifically, we manually remove important information randomly during the table condensation step (at the root) and evaluate

the performance. The experimental results show that even when a certain amount of important information is removed, the performance of Tree-of-Table does not significantly degrade and still achieves relatively correct results, indicating the robustness of our method.

### A.1.5 CASE STUDY

To more intuitively verify the superiority of our method, we have also included specific case comparisons in Table 11, which encompass two typical complex logic problems: precise calculation and statistics.

### A.2 PROMPT TEMPLATE

The main structure of prompt template of Tree-of-Table is shown in Figure 5.

### A.3 THEORETICAL ADVANTAGES OF HIERARCHICAL TREE-BASED ARCHITECTURE

Previous work has shown that tree-based structures exhibit significant advantages in LLM reasoning, as demonstrated in (Yao et al., 2023b; Zhou et al., 2022). In addition, earlier works also demonstrate from multiple perspectives that hierarchical tree-based architecture is superior to linear architecture, including but not limited to the following points:

- Divide and Conquer Strategy (Bentley, 1975): Tree-based architectures leverage the divide-and-conquer approach, which is a well-established algorithmic paradigm. This strategy breaks a problem into smaller subproblems, solves each subproblem independently, and combines their results. This can lead to more efficient processing, especially in complex, large-scale problems.
- Scalability (Blumofe & Leiserson, 1999): Trees can handle larger and more complex datasets more efficiently than linear structures. As data grows, the depth of a tree increases logarithmically rather than linearly, allowing for more scalable processing.
- Improved Decision Making (Loh, 2011): In decision-making processes, tree structures can better model decision paths and outcomes, providing clearer insights into the reasoning behind decisions.
- Cognitive Alignment (Newell, 1972; Newell et al., 1959; Miller, 1956): Human reasoning often aligns more closely with hierarchical structures, which can make tree-based models more intuitive and easier to interpret.

### A.4 REPHRASE THE PROCESS OF CHAIN-OF-TABLE

Overall, Chain-of-Table processes table understanding based on the Chain-of-Thought LLM reasoning approach, which employs an intuitive linear thought chain to decompose the problem. At each step of decomposition, it selects operations from a predefined operations pool and generates intermediate results for table processing. However, when dealing with complex, multi-branch logic, this linear approach can produce lengthy and disorganized thought processes, making table reasoning chaotic and prone to errors. Therefore, this linear approach is generally suited for relatively simple problems and smaller tables.

In contrast, our proposed Tree-of-Table method is divided into three main parts: (1) Table Condensation (Sec. 3.3): Given the input table, which contains many large-scale relevant tables connected via foreign keys, we first employ LLMs to condense the table. This helps recall the sub-tables pertinent to the query and eliminate extraneous information. (2) Table-Tree Construction (Sec. 3.4): Following the preprocessing steps, we instruct the LLM to dynamically generate child nodes in the tree. Each generation of child nodes relies solely on their multi-level parent nodes, without needing reference to uncle nodes. We then iteratively construct child nodes level by level until we reach the leaf nodes at the bottom of the tree. (3) Table-Tree Execution (Sec. 3.5): Finally, after constructing the Table-Tree, we consider it a proxy task for the entire table understanding process. The traversal and execution of operations across this tree enable a seamless reasoning process.

In summary, the differences between the two methods are as illustrated in Sec. 3.6.

Table 11: Case Study.

| Question | Chain-of-Table | Tree-of-Table | Groundtruth |
|---|---|---|---|
| "What was the growth rate of the total amount of loans across all accounts for a male client between 1996 and 1997?" | 0 | 25.30 | 25.30 |
| "Consider the average difference between K-12 enrollment and 15-17 enrollment of schools that are locally funded, list the names and DOC type of schools which has a difference above this average" | Incomplete list | Mountain Oaks(00), Castle Rock(00), Charter Community School Home Study Academy(00), Clovis Online Charter(54), Washington Elementary(52), ...... | Mountain Oaks(00), Castle Rock(00), Charter Community School Home Study Academy(00), Clovis Online Charter(54), Washington Elementary(52), ...... |
| "What is the e-mail address of the administrator of the school located in the San Bernardino county, District of San Bernardino City Unified that opened between 1/1/2009 to 12/31/2010 whose school types are public Intermediate/Middle Schools and Unified Scools?" | www.realjourney.org | a.lucero@realjourney.org | a.lucero@realjourney.org |
| "For the branch which located in the south Bohemia with biggest number of inhabitants, what is the percentage of the male clients?" | 40 | 44.26 | 44.26 |

## Task Instruction ##
You are an information and database analysis expert. When faced with a specific database with some tables and a question, please use logical analysis methods to rigorously output the answer from the provided information, given the instruction.

## Guidelines ##
Before answering the question, you will be presented with some /*Database tables*/, /*Database schema*/ description, a knowledge /*Evidence*/ and the /*Question*/. You will rely on the provided information, utilizing methods of logical analysis, to answer the question using only the provided materials. Aiming to provide an accurate answer.

/*Instruction*/
Perform Table-Tree Construction based on the operation pool and then Execute the Table-Tree.

[Table-Tree Construction]:

- Thought Decomposition: You should generate a Tree-like thought chains for the question and tables. Following by the "divide and conquer" philosophy, each generation of child nodes relies exclusively on the information from their parent nodes, without the need for uncle nodes. The intermediate nodes represent a subprocess that can be further decomposed.

- Operation Sample: The leaf nodes are the actionable endpoints of the tree where specific operations such as data retrieval, calculations can execute based on the parameters defined by their parent nodes. You will sample the operations from the Operation Pool

[Table-Tree Execution]: Execute the table-tree by depth-first searching

/*Operation Pool*/
...... // omission of operation pool, following Chain-of-Table

## Some In-context examples ##

/*Database tables*/
The condensated table is:
...... // omission of table content

/*Database schema*/
Customers: [
(CustomerID, customer ID. Value example: [3, 5, 6].),
(Segment, consumption type. Value example: ['SME', 'LAM', 'KAM'].),
(Currency, currency. Value example: ['EUR', 'CZK'].)
] (foreign key: CustomerID, Segment)

Gasstations: [
(GasStationID, gas station ID. Value example: [44, 45, 60].),
(ChainID, chain ID. Value example: [13, 6, 10].),
(Country, country name. Value example: ['CZE']),
(Segment, consumption type. Value example: ['SME', 'LAM', 'KAM'].)
] (foreign key: CustomerID, Segment, GasStationID)

Products: [
(ProductID, product ID. Value example: [1, 3, 5].),
(Description, product description. Value example: ['Special', 'Protraviny'].),
] (foreign key: ProductID)

Transactions_1k: [
(TransactionID,  transaction ID. Value example: [11, 13, 16].),
(Date, date. Value example: [2012-08-24].),
(Time, time. Value example: [09:41:00].),
(CustomerID, customer ID. Value example: [3, 5, 6].),
(CardID, card ID. Value example: [486621].),
(GasStationID, gas station ID. Value example: [44, 45, 60].),
(ProductID, product ID. Value example: [1, 3, 5].),
(Amount, amount. Value example: [28, 18].),
(Price, price. Value example: [2, 8].),
] (foreign key: CustomerID, ProductID, GasStationID)

Yearmonth: [
(CustomerID, customer ID. Value example: [3, 5, 6].),
(Date, date. Value example: [2012-08-24].),
(Consumption, consumption. Value example: [528.3])
] (foreign key: CustomerID)

/*Question*/
Which of the three segments—SME, LAM and KAM—has the biggest and lowest percentage increases in consumption paid in EUR between 2012 and 2013?

/*Evidence*/
Increase or Decrease = consumption for 2013 - consumption for 2012; Percentage of Increase = (Increase or Decrease / consumption for 2013) * 100%; Between 2012 And 2013 can be represented by Between 201201 And 201312; First 4 strings of Date represents the year.

/*Output of Table-Tree */
"i" is intermediate node, "l" is leaf node
(Tree-1 Chain): Increase Percentage of Consumption for SME (i) → Consumption of SME (i) → EUR (i) → 2012 (i)→ [sum sum … cumulative] (l) → 2013 (i) → [sum sum … cumulative] (l) → others (i) → EUR (i) → Not EUR (i) → Consumption of SME (i) → Percentage of Increase of SME (i) → Increase (i) → subtract (l) → Percentage (i) → divide (l) → Percentage of Increase of SME (i) → Increase Percentage of Consumption for SME (i)
→ (Tree-2 Chain):  ... // omission of Tree-2 Chain
→ (Tree-3 Chain): …. // omission of Tree-3 Chain
→ (Tree-4 Chain): Compare biggest and lowest (i)→ sort_by (l)
The 4 chains are execute one by one

/*Answer*/
KAM has the biggest percentage increases. SME has the lowest percentage increase.

…….. # provide more examples

……. # omission of operation pool, could refer to Chain-of-Table

## Query ##
/*Database tables*/
…… // omission of table content

/*Database schema*/
[
  (sname, school name. Value examples: [Making Waves Academy].),
  (cname, county name. Value examples: [Contra Costa].),
  (NumTstTakr, Number of Test Takers in this school. Value examples: [73].),
]
……

/*Question*/
Which school in Contra Costa has the highest number of test takers?

/*Evidence*/
…… // omission

/*Output of Table-Tree */

/*Answer*/

Figure 5: The main structure of prompt template of Tree-of-Table.

