# OpenReview forum: "Tree-of-Table: Unleashing the Power of LLMs for Enhanced Large-Scale Table Understanding"
_ICLR.cc/2025/Conference — Submitted to ICLR 2025_

### Official Review · Reviewer_MiAo · 2024-10-22

**Soundness:** 3
**Presentation:** 3
**Contribution:** 2
**Rating:** 5
**Confidence:** 4

**Summary:**

The paper introduces the "Tree-of-Table" method, designed to improve the reasoning abilities of large language models (LLMs) when dealing with large and complex tabular data. The approach involves two main steps: Table Condensation and Decomposition, which simplifies and organizes the data, and Hierarchical Table-Tree Construction, creating a structured representation that benefits systematic reasoning. This method enhances the efficiency and generalization capabilities of LLMs, demonstrated through superior performance on datasets like WikiTQ, TableFact, FeTaQA, and BIRD. This study advances LLM methods for parsing and understanding extensive tabular datasets, setting new benchmarks in handling complex table-based information.

**Strengths:**

* The idea of using tree as a roadmap to guide the LLMs through table(s) is sound but seems like an adaptation from other work to table tasks. It's not very novel to consider increment LLMs using chain-of-thoughts to tree-of-thoughts, which have been already verified on other domains, like graphs.

**Weaknesses:**

W: I'm unclear about the logic in the introduction from lines 94 to 103, specifically why the Chain-of-table is limited to smaller tables. What does 'smaller' refer to exactly? Is it generally about the number of rows/columns or the total number of tokens across all cells?

W: The performance of the proposed tree-of-table is limited, especially compared with the chain-of-table referred to in Table 1 and 2. Additionally, it's not very easy to differentiate the contribution of tree-of-table and chain-of-table. The authors should consider providing a critical analysis about the difference in the introduction or method section, and show how this improvement can account for the performance enhancement.

Several relevant papers should be considered in references:

* Large Language Models are few(1)-shot Table Reasoners
* StructGPT: A General Framework for Large Language Model to Reason over Structured Data
* TAP4LLM: Table Provider on Sampling, Augmenting, and Packing Semi-structured Data for Large Language Model Reasoning
* Table Meets LLM: Can Large Language Models Understand Structured Table Data? A Benchmark and Empirical Study
* TableRAG: Million-Token Table Understanding with Language Models

**Questions:**

Q1: I'm unclear about the logic in the introduction from lines 94 to 103, specifically why the Chain-of-table is limited to smaller tables. What does 'smaller' refer to exactly? Is it generally about the number of rows/columns or the total number of tokens across all cells or like a database: the number of tables linked with keys?

Q2: what is the OP pool mentioned in Figure 1? I suggest the authors to rephrase the process of chain-of-table in case not all readers are familiar with this method. Additionally, the authors should also describe the difference between chain-of-table and proposed tree-of-table explicitly.

**Details Of Ethics Concerns:**

**In the review of the manuscript, I observed that lines 138-141, where the authors state "our proposed chain-of-table approach innovates by generalizing a broader set of ....", may validate the double-blind requirement of the ICLR. The mentioned chain-of-table, which is another paper, might reveal the authors' identities.**

I recommend that the ACs evaluate this situation to decide if the paper warrants a **desk reject**.

Reference:

Chain-of-table: Evolving tables in the reasoning chain for table understanding (https://scholar.google.com/scholar_url?url=https://arxiv.org/pdf/2401.04398&hl=en&sa=T&oi=gsr-r-gga&ct=res&cd=0&d=13815199908318046768&ei=7D0XZ8LsJsDBy9YP75aH4Qc&scisig=AFWwaeauOr6b6P9gI77Y9IO8x6Tf)

---

> ### Author Response · Authors · 2024-11-13
> **Clarification on Ethics Concerns**
>
> We would like to quickly clarify the identity leakage concern before submitting the formal rebuttal. The issue you mentioned in Lines 138-141 is actually a typo; it should be “our proposed tree-of-table” rather than “chain-of-table”, indicating our proposed method in this paper. Therefore, it does NOT reveal any information about the authors’ identity. We apologize for the typo and any confusion it may have caused, and we will correct it in the next version.

---

> ### Author Response · Authors · 2024-11-23
> **Response to the comments of Reviewer MiAo**
>
> Thank you for recognizing the significance of Tree-of-Table. Below, we provide explanations addressing the concerns you have raised.
>
> ## For Weakness and Question 1
>
> *“why the Chain-of-table is limited to smaller tables”*
>
> Overall, Chain-of-Table processes table understanding based on the Chain-of-Thought LLM reasoning approach, which employs an intuitive linear thought chain to decompose the problem. At each step of decomposition, it selects operations from a predefined operations pool and generates intermediate results for table processing. However, when dealing with large tables accompanied by complex, multi-branch logic, this linear approach can produce lengthy and disorganized thought processes, making table reasoning chaotic and prone to errors. Therefore, this linear approach is generally suited for relatively simple problems and smaller tables. In contrast, the hierarchical tree structure of Tree-of-Table is naturally suited for handling complex logical problems [1, 2, 3] and provides a natural and scalable way of reasoning while operating within the context limits of LLMs.
>
>
> *“What does 'smaller' refer to exactly?”*
>
> In previous work [4,5,6,7], the size of a table typically refers to the number of rows. In our work, we also follow the widely used setting in previous works, considering tables with more than 30 rows as large-scale tables. Additionally, we have also taken into account the recently published BIRD dataset, which is very large and consists of 549K rows per database.
>
> *"Performance"*
>
> In the below table, we present a comparison on performance and computation between our proposed Tree-of-Table and the previous SOTA method Chain-of-Table. Compared to Chain-of-Table, our approach requires only about 20% more computation, yet achieves a significant improvement of 3.58 in BLEU score on such a large-scale BIRD table-based dataset, with even less resource consumption. This demonstrates the significant advantages of our method.
>
> |     Method     |  BLEU | Time cost | Generated Samples |
> |:--------------:|:-----:|:---------:|:-----------------:|
> | Chain-of-Table | 12.12 |    5.7    |        120        |
> |  Tree-of-Table | 15.70 |    7.8    |         90        |
>
> *“Several relevant papers should be considered in references"*
>
> Thank you for your suggestion. We have incorporated these references into the revised version.

---

> > ### Author Response · Authors · 2024-11-23
> > **Response to the comments of Reviewer MiAo**
> >
> > ## For Weakness and Question 2
> >
> > *“what is the OP pool mentioned in Figure 1?”*
> >
> > As mentioned in Section 3.4.2, OP Pool refers to the operations pool, which represents the set of operations from which the LLM dynamically selects those relevant to table-based reasoning, such as select, group_by, sort_by, and others. In selecting the operations pool, we based our choices on existing work [4] and selected the most frequently used table operations from the resource at [8].
> >
> > *“rephrase the process of chain-of-table” and “ the difference between chain-of-table and proposed tree-of-table”*
> >
> > Following your suggestion, we have refined the formulation of Chain-of-Table and Tree-of-Table to provide you with a clearer understanding of their differences,  in Sec. 3.6 and Sec. A.4 of the revised paper version.
> > Overall, Chain-of-Table processes table understanding based on the Chain-of-Thought LLM reasoning approach, which employs an intuitive linear thought chain to decompose the problem. At each step of decomposition, it selects operations from a predefined operations pool and generates intermediate results for table processing. However, when dealing with complex, multi-branch logic, this linear approach can produce lengthy and disorganized thought processes, making table reasoning chaotic and prone to errors. Therefore, this linear approach is generally suited for relatively simple problems and smaller tables.
> > In contrast, our proposed Tree-of-Table method is divided into three main parts:
> > - Table Condensation (Sec. 3.3): Given the input table, which contains many large-scale relevant tables connected via foreign keys, we first employ LLMs to condense the table. This helps recall the sub-tables pertinent to the query and eliminate extraneous information.
> > - Table-Tree Construction (Sec. 3.4): Following the preprocessing steps, we instruct the LLM to dynamically generate child nodes in the tree. Each generation of child nodes relies solely on their multi-level parent nodes, without needing reference to uncle nodes. We then iteratively construct child nodes level by level until we reach the leaf nodes at the bottom of the tree.
> > - Table-Tree Execution (Sec. 3.5): Finally, after constructing the Table-Tree, we consider it a proxy task for the entire table understanding process. The traversal and execution of operations across this tree enable a seamless reasoning process.
> >
> > In summary, the differences between the two methods are as follows:
> > - Planning Style: Tree-of-Table uses a hierarchical, tree-based thought decomposition approach, while Chain-of-Table uses a linear thought decomposition approach.
> > - Execution Strategy: Notably, Chain-of-Table processes the entire chain of history for each dynamic planning step and is shown to be effective for relatively small tables. However, as tables grow larger and questions become more complex, maintaining the complete thought chain becomes cumbersome, ultimately decreasing the efficiency of the model. Our Tree-of-Table method addresses this by embracing the tree's inherent "divide and conquer" philosophy to construct a Table-Tree. Each generation of child nodes relies exclusively on the information from their multi-level parent nodes, without the need for uncle nodes. By this way, we significantly reduce the historical chain's length on which each node's dynamic planning relies, to less than the depth of the tree, thus considerably simplifying the generation process at each level.
> > - Data Preprocessing: Chain-of-Table does not preprocess complex tables with rich foreign key connections, whereas Tree-of-Table includes a Table Condensation step to handle such complexities.

---

> ### Author Response · Authors · 2024-11-23
> **Response to the comments of Reviewer MiAo**
>
> ### Clarification on Ethics Concerns
>
> Sorry for this! The issue you mentioned in Lines 138-141 is actually a typo; it should be “our proposed tree-of-table” rather than “chain-of-table”, indicating our proposed method in this paper. Therefore, it does NOT reveal any information about the authors’ identity. We apologize for the typo and any confusion it may have caused, and we have corrected it in the revised version.
>
> [1] Yao S, Yu D, Zhao J, et al. Tree of thoughts: Deliberate problem solving with large language models[J]. Advances in Neural Information Processing Systems, 2023
>
> [2] Denny Zhou, Nathanael Schärli, Le Hou, Jason Wei, Nathan Scales, Xuezhi Wang, Dale Schuurmans, Claire Cui, Olivier Bousquet, Quoc Le, et al. Least-to-most prompting enables complex reasoning in large language models. arXiv preprint arXiv:2205.10625, 2022.
>
> [3] Jon Louis Bentley. Multidimensional binary search trees used for associative searching. Communications of the ACM, 18(9):509–517, 1975.
>
> [4] Wang, Zilong, et al. Chain-of-table: Evolving tables in the reasoning chain for table understanding. ICLR 2024.
>
> [5] Nahid M M H, Rafiei D. Tabsqlify: Enhancing reasoning capabilities of llms through table decomposition. NAACL 2024
>
> [6] Chen W. Large language models are few-shot table reasoners. EACL 2023
>
> [7] Ye Y, Hui B, Yang M, et al. Large language models are versatile decomposers: Decomposing evidence and questions for table-based reasoning. SIGIR 2023
>
> [8] Bytescout. "https://bytescout.com/blog/20-important-sql-queries.html"
>
> ------
>
> Thank you for your valuable comments.  We hope the responses provided above sufficiently address your concerns. Shall you have any further questions, we are more than happy to address them.

---

> > ### Comment · Reviewer_MiAo · 2024-11-26
> >
> > Thanks for the detailed response. The idea of using a tree as a roadmap to guide the LLMs is sound. However, the new results concerning the effectiveness and efficiency of the Chain-of-table and Tree-of-table raise some concerns. It seems that the Chain-of-thoughts might not perform as bad on larger tables as initially claimed in both manuscript and author's responses, instead it stills achieves a quite similar performance compared with the proposed method even requires less time. This raises further concerns that the new method might overcomplicated the systems and the gained improvements is limited. Given this fundamental issue, it's difficult for me to provide a positive rating. Nevertheless, I appreciate the additional information from the authors. I will maintain my current rating and look forward to future discussions with other reviewers and ACs.

---

> ### Author Response · Authors · 2024-11-27
>
> Thank you for your feedback. In our work, regarding the issue of efficiency, we evaluate it from two aspects:
>
> - Resource Consumption: Following the evaluation metric of Chain-of-Table, we use the number of generated samples in Table 5 to evaluate the resource consumption, in which Tree-of-Table has a smaller value on this metric compared to Chain-of-Table. This demonstrates that Tree-of-Table is more resource-friendly on such a large-scale dataset like BIRD (549K rows per database). This is because, during the construction of the Table-Tree, we limit the tree depth and breadth, which helps prevent the LLM from engaging in meaningless, excessive thoughts that could lead to repetitive or erroneous reasoning processes.
>
> - Latency/Time: Tree-of-Table does have more latency compared to Chain-of-Table. This is due to the Depth-First Search Execution method employed by Tree-of-Table, which introduces some time overhead for searching and backtracking.
>
> In summary, Tree-of-Table has relatively lower resource consumption and a reasonable additional latency compared to Chain-of-Table. Considering that we achieve a 3.58 improvement in BLEU on such a large-scale dataset like BIRD (549K rows per database), we believe Tree-of-Table is a very practical method. Additionally, as shown in Figure 4(a), Tree-of-Table also demonstrates strong generalization across datasets of different scales. Based on the above analysis, we refine the experimental results in the previous response.
>
> We hope the above response addresses your concerns.
>
> Best wishes to you
>
> Paper Authors

---

### Official Review · Reviewer_etFU · 2024-10-31

**Soundness:** 3
**Presentation:** 3
**Contribution:** 4
**Rating:** 6
**Confidence:** 4

**Summary:**

This paper introduces "Tree-of-Table" to enhance LLMs' ability to understand and reason with large-scale tabular data. Key contributions include a framework with table condensation that distills relevant information from large tables, table-tree construction, which organizes reasoning steps into a hierarchical tree structure, and then table-tree execution which systematically processes the tree through DFS. The structure breaks down complex table understanding tasks into manageable sub-problems to allow more efficient processing compared to linear chain approaches. Their experiments demonstrated enhanced performance over existing methods on large-scale tables across multiple datasets including wikiTQ, TableFact, FeTaQA, and BIRD.

**Strengths:**

This paper introduced a tree structure for handling tabular data which is different from the traditional linear chain-of-thought and more recent chain-of-table methods. What I like in particular is how they combined table condensation with tree decomposition - the authors seem to have thought carefully about how humans break down complex problems, and they've used the insights and built this into their approach. The experimental work is solid. They tested their method on several different datasets (WikiTQ, TableFact, FeTaQA, and BIRD), which gives us confidence in the results. The numbers are impressive - they're getting better performance than existing methods, especially on BIRD which has those really large tables that are typically hard to handle. I was particularly convinced by their ablation studies.

The paper is easy to follow. The figures really help explain what's going on - Figure 1 does a great job showing how their approach differs from previous methods. They've managed to explain some pretty complex technical stuff without making it too dense. That said, they could have made some of the implementation details clearer. In terms of impact, this work matters because large-scale table understanding is a real problem that comes up all the time in practical applications. Their method shows promise for handling tables in finance, healthcare, and other fields where you often deal with complex tabular data. The performance improvements they're showing are consistent across different LLMs, and across various table understanding datasets. What stands out most to me is that they've taken a practical problem (with demonstrated efficiency improvement against other methods) that lots of people struggle with and come up with a solution that actually works better than what we had before. The evidence is there in their results, and they've explained their approach well enough that others could build on it.

**Weaknesses:**

1. The theoretical foundation needs more work. While the tree-based approach shows good empirical results, there's limited analysis of why it works better than linear chains.

2. Some key experimental details are missing or unclear: They don't specify how they chose parameters like MAXDegree and MAXDepth for the Table-Tree. These seem pretty important for the method's performance. Alsothe computational overhead of building and traversing the tree structure wasn't properly analyzed, for example - memory requirements for storing intermediate results at tree nodes and overall computational complexity compared to simpler approaches

3. The ablation studies could go deeper. There's no clear analysis of how the tree structure's depth affects accuracy. The comparison with Chain-of-Table focuses mainly on final accuracy, but doesn't explore cases where their method might perform worse

**Questions:**

1. The authors should explain the theoretical advantages of their hierarchical decomposition - when does it work better and why? This would help us understand the method's limitations and where it might fail.
2. There's no discussion of error propagation through the tree structure. In a tree structure, errors at higher levels will propagate down through all child nodes. For example, if the table condensation step (at the root) removes important information, or if an early operation in the tree is incorrect, how does this affect the final result? The paper shows good overall accuracy but doesn't analyze these failure cases.
3. Related to above, how sensitive is your method to the quality of the initial table condensation step? What happens if crucial information is accidentally filtered out?
4. What is the complete set of operations in the operation pool? How were these operations selected and validated?
5. Have you analyzed cases where Tree-of-Table performs worse than Chain-of-Table? This would be valuable for understanding the method's limitations.

---

> ### Author Response · Authors · 2024-11-23
> **Response to the comments of Reviewer etFU**
>
> Thank you for your praise and appreciation of our work, including your remarks that "the experimental work is solid", "the numbers are impressive", "easy to follow", "practical" and so on. Encouraged by your comments, we are pleased to respond to your valuable suggestions to further improve our work.
>
> ## For Weakness 1 and Question 1
>
> Previous work has shown that tree-based structures exhibit significant advantages in LLM reasoning, as demonstrated in [1,2]. In addition, earlier works have demonstrated from multiple perspectives that hierarchical tree-based architecture is superior to linear architecture, including but not limited to the following points:
>
> 1.	Divide and Conquer Strategy [3]: Tree-based architectures leverage the divide-and-conquer approach, which is a well-established algorithmic paradigm. This strategy breaks a problem into smaller subproblems, solves each subproblem independently, and combines their results. This can lead to more efficient processing, especially in complex, large-scale problems.
>
> 2.	Scalability [4]: Trees can handle larger and more complex datasets more efficiently than linear structures. As data grows, the depth of a tree increases logarithmically rather than linearly, allowing for more scalable processing.
>
> 3.	Improved Decision Making [5]: In decision-making processes, tree structures can better model decision paths and outcomes, providing clearer insights into the reasoning behind decisions.
>
> 4.	Cognitive Alignment [6,7,8]: Human reasoning often aligns more closely with hierarchical structures, which can make tree-based models more intuitive and easier to interpret.
>
> We have also included the above discussions in the appendix (Sec. A.3) of the revised paper version.
>
> ## For Weakness 2 and Weakness 3
>
> Following your suggestion, we have added the following experiments:
>
> 1.	Selection of MAXDegree and MAXDepth for the Table-Tree: In our experiments, we initially conduct preliminary experiments to roughly determine their value ranges. We then perform detailed hyperparameter experiments to identify the optimal values. The ablation study results on WikiTQ are listed in the table below. It can be seen that our proposed method is relatively robust with respect to these two hyperparameters.
>
> |     MAXDepth    |     6       |     8       |     10       |
> |-----------------|--------------|--------------|--------------|
> |     Accuracy(%) |     59.96    |     61.11    |     60.47    |
>
> |     MAXDegree    |     3        |     4        |     5        |
> |------------------|--------------|--------------|--------------|
> |     Accuracy(%)      |     60.03    |     61.11    |     60.91    |
>
> 2. Memory Efficiency of Tree-of-Table: In fact, Tree-of-Table is a training-free, prompt-based method that requires no additional data, making it very memory-efficient. In our analysis, we find that the tree-of-table structure increased memory cost by less than 5% compared to the chain-of-table structure.
>
>
> ## For Question 2 and Question 3
>
> To address your concern regarding error propagation, we have added the following study. Specifically, we manually remove important information randomly during the table condensation step (at the root) and evaluate the performance. The experimental results  on WikiTQ below show that even when a certain amount of important information is removed, the performance of Tree-of-Table does not significantly degrade and still achieves relatively correct results, indicating the robustness of our method.
>
> |     Remove Percent(%)    |     5        |     8        |     10       |     12       |     15       |
> |--------------------------|--------------|--------------|--------------|--------------|--------------|
> |     Accuracy(%)            |     61.09    |     61.00    |     60.87    |     60.22    |     59.45    |

---

> > ### Comment · Reviewer_etFU · 2024-11-26
> >
> > I thank authors for their response which addressed most of my questions. I would keep the score since it was already positive.

---

> > > ### Author Response · Authors · 2024-11-26
> > >
> > > Thanks very much for your  feedback! We are glad that our discussions addressed your concerns.
> > >
> > > Best wishes to you
> > >
> > > Paper Authors

---

> ### Author Response · Authors · 2024-11-23
> **Response to the comments of Reviewer etFU**
>
> ## For Question 4
>
> In selecting the operation pool, we follow the method of existing work [9], choosing the most frequently used table operations from the resource at [10]. This selection is found to be effective in practical, even on the large-scale dataset like BIRD.
>
> ## For Question 5
>
> Indeed, we do analyze the worse cases in our experiments, which typically stem from excessively unreasonable tree depth. An abnormally large depth can lead to reduced efficiency and chaotic thoughts in the Tree-of-Table, while an abnormally small depth severely limits the LLM's reasoning potential. Both of these extreme settings can result in a certain number of worse cases.
>
> [1] Yao S, Yu D, Zhao J, et al. Tree of thoughts: Deliberate problem solving with large language models[J]. Advances in Neural Information Processing Systems, 2023
>
> [2] Denny Zhou, Nathanael Schärli, Le Hou, Jason Wei, Nathan Scales, Xuezhi Wang, Dale Schuurmans, Claire Cui, Olivier Bousquet, Quoc Le, et al. Least-to-most prompting enables complex reasoning in large language models. arXiv preprint arXiv:2205.10625, 2022.
>
> [3] Jon Louis Bentley. Multidimensional binary search trees used for associative searching. Communications of the ACM, 18(9):509–517, 1975.
>
> [4] Robert D Blumofe and Charles E Leiserson. Scheduling multithreaded computations by work stealing. Journal of the ACM (JACM), 46(5):720–748, 1999.
>
> [5] Wei-Yin Loh. Classification and regression trees. Wiley interdisciplinary reviews: data mining and
> knowledge discovery, 1(1):14–23, 2011.
>
> [6] llen Newell. Human problem solving. Upper Saddle River/Prentive Hall, 1972.
>
> [7] Allen Newell, John C Shaw, and Herbert A Simon. Report on a general problem solving program. In IFIP congress, volume 256, pp. 64. Pittsburgh, PA, 1959.
>
> [8] George A Miller. The magical number seven, plus or minus two: Some limits on our capacity for processing information. Psychological review, 63(2):81, 1956.
>
> [9] Wang, Zilong, et al. Chain-of-table: Evolving tables in the reasoning chain for table understanding. ICLR 2024.
>
> [10] Bytescout. "https://bytescout.com/blog/20-important-sql-queries.html"
>
>
> -------
>
> Thank you for your appreciation of Tree-of-Table. We have improved our paper based on your insightful comments.

---

### Official Review · Reviewer_SjDx · 2024-11-03

**Soundness:** 3
**Presentation:** 3
**Contribution:** 2
**Rating:** 5
**Confidence:** 4

**Summary:**

This paper proposes a tree-of-table method, which generates the tree thoughts to improve the LLMs' reasoning ability on large-size tables. The experiments are conducted on multiple datasets such as WikiTQ, TableFact, FeTaQA and BIRD and show the better performance than baselines.

**Strengths:**

1. The experimental results show the proposed tree-of-table method leads to better performance than baselines.
2. The experiments are conducted on multiple table-based datasets and show the effectiveness of the proposed method.

**Weaknesses:**

1. Deriving such a huge tree for table QA can raise the efficiency concern.
2. It would be better to show some cases in which tree-of-table can handle better than chain-of-table.

**Questions:**

1. What types of queries and tables can mainly benefit from tree-of-table rather than chain-of-table?
2. How easy will it go into a dead loop during the derivation of the trees?

---

> ### Author Response · Authors · 2024-11-23
> **Response to the comments of  Reviewer SjDx**
>
> We sincerely thank you for your valuable comments and acknowledging the effectiveness of our methods on multiple table-based datasets. In response to the questions and suggestions you raised, we would like to offer the following replies.
>
> ## For Weakness 1
>
> Yes, this is indeed a worthwhile consideration! In the previous submitted paper, we have analyzed the efficiency of Tree-of-Table from multiple perspectives, including the number of generated samples and overall time costs. This is consistent with the evaluation method used for efficiency analysis in Chain-of-Table. These analyses can be found in Section 4.3 "Efficiency Analysis", Table 5, Section A.1.1 "Ablation Study on Time Cost", Table 6 of main paper. For your convenience, in the below table, we present a comparison on performance and computation between our proposed Tree-of-Table and the previous SOTA method Chain-of-Table. Compared to Chain-of-Table, our approach requires only about 20% more lantency, yet achieves a significant improvement of 3.58 in BLEU score on such a large-scale BIRD table-based dataset, with even less resource consumption. This demonstrates the significant advantages of our method.
>
>
> |     Method     |  BLEU | Time cost | Generated Samples |
> |:--------------:|:-----:|:---------:|:-----------------:|
> | Chain-of-Table | 12.12 |    5.7    |        120        |
> |  Tree-of-Table | 15.70 |    7.8    |         90        |
>
> ## For Weakness 2 and Question 1:
>
> Overall, our Tree-of-Table has advantages over Chain-of-Table in the following types of queries and tables:
>
> 1. Complex Query Logic: Tree-of-Table excels when queries involve complex logic, typically requiring multi-layered nested table tracing, filtering, comparison, and computation. As we discussed in Section 3.4.2 (the previous submitted version) and Sec. 3.6 (the new revised version), the hierarchical construction and execution of Tree-of-Table are particularly advantageous.
>
> 2. Large-Scale Tables: Tree-of-Table demonstrates superior performance on large-scale tables, especially when there are many rows. As mentioned in Section 4.3 "Generalization Ability under Different Table Sizes" and Figure 4a, Tree-of-Table exhibits stronger generalization ability relative to table size.
>
> Following your suggestion, we have included the case studies in appendix of the revised paper version (Sec. A.1.5) that demonstrate the superior performance of our proposed Tree-of-Table compared to Chain-of-Table.
>
> ## For Question 2
>
> Nice insight! We indeed encountered the situation you mentioned in our initial design. To address this, in our method, we have incorporated prompts specifically instructing the LLM to keep the decomposition within the limits of the input context and prevent the dead loop. This approach helps prevent the LLM from engaging in meaningless, excessive thoughts that could lead to repetitive or erroneous reasoning processes. Specifically, as we mentioned in Section 3.4.1, we set a maximum value for the depth and degree, denoted as "MAXDepth" and “MAXDegree”. Therefore, in our experiments, the derivation of the trees does not result in a dead loop as concerned by you.
>
> Thank you for your valuable comments. We hope that the above responses can address your concerns. Shall you have any further questions, we are more than happy to address them.

---

### Meta-Review · Area_Chair_GZUT · 2024-12-23

**Metareview:**

This paper presents "Tree-of-Table", a method to improve LLMs' reasoning capabilities over complex tabular data. The proposed method decomposes a reasoning task into several hierarchical steps with a tree structure. Experiments on four datasets demonstrate the effectiveness of the proposed approach.

As raised by Reviewer MiAo ([link](https://openreview.net/forum?id=Yv8FrCY87H&noteId=mCvFHiOM50)), the major issue is that the advantages of the proposed method might be limited compared to the existing chain-of-tree approach, as the latter “achieves a quite similar performance compared with the proposed method even requires less time”, given this fundamental issue, the decision is rejection.

**Additional Comments On Reviewer Discussion:**

There are additional minor issues on clarity and lack of certain ablations, which were addressed during the rebuttal period.

---

### Decision · Program_Chairs · 2025-01-22

Reject